# Physics-Informed Machine Learning under Climate Domain Shift: PDE-Free Physics Regularisation for Cloud Prediction

## Abstract

We study out-of-distribution generalisation in geophysical prediction and propose CC-PINN, a physics-informed multi-layer perceptron (MLP) that encodes the Clausius–Clapeyron thermodynamic relation as a gradient-based regularisation term. Unlike prior PINNs, CC-PINN requires no explicit governing-equation. CC-PINN introduces a lightweight constraint on humidity-temperature consistency without altering network architecture. Trained on atmospheric re-analysis data (temperature, pressure, relative humidity, specific humidity, vertical velocity) using modest computational resources, CC-PINN matches a capacity-matched MLP in-distribution and improves out-of-distribution performance. CC-PINN achieves a 12.6% reduction in global area-weighted RMSE over a capacity-matched MLP baseline. Under a stringent covariate-shift test - training only on the polar latitudes - CC-PINN reduces tropical area-weighted root mean squared error (RMSE) by 23.6% relative to the baseline, while maintaining in-distribution parity. Ablations show the performance gains are substantially attenuated when the physics term is removed, highlighting the role of targeted domain knowledge inclusion in improving extrapolation. These findings suggest that compact, domain-motivated regularisation can deliver robust generalisation in scientific ML tasks.

## 1 Introduction

Out-of-distribution (OOD) generalisation remains a fundamental challenge in machine learning, where models, often trained under one data regime are routinely deployed in others, (Koh et al., 2021). In climate modelling, these shifts are not rare anomalies but the norm: evolving climate states can produce input distributions that differ significantly from historical training data, (Beucler et al., 2024). Standard neural networks excel at in-distribution prediction but frequently overfit to spurious correlations, resulting in degraded performance when faced with new regimes, (Arjovsky et al., 2020).

Cloud fraction prediction is a critical example of this problem. Global circulation models (GCMs) cannot explicitly resolve the small-scale processes that govern cloud formation, so they rely on parameterisations—empirical or semi-empirical approximations that introduce uncertainty into climate projections (Stephens, 2005; Bony et al., 2006; Smith, 1990a; Tiedtke, 1993). Recent work has explored replacing or augmenting these schemes with machine-learning surrogates trained on reanalysis or high-resolution simulation data (Rasp et al., 2018; Brenowitz & Bretherton, 2019). While such models can achieve low error in familiar conditions, they often fail to maintain accuracy when extrapolating to different climatic zones or unseen meteorological states (Dueben & Bauer, 2018; Beucler et al., 2024). In this work we do not seek to design a new state-of-the-art cloud-fraction parameterisation. Instead, we use a simple, capacity-matched MLP as a controlled testbed to study how a Clausius–Clapeyron–motivated constraint affects robustness under regime shifts.

We address this gap by introducing CC-PINN, a physics-informed MLP that incorporates the Clausius–Clapeyron (CC) thermodynamic relation directly into the loss function as a gradient-based regularisation term (Raissi et al., 2019). The CC relation governs the dependence of saturation vapour pressure on temperature, a fundamental driver of cloud formation (Wallace & Hobbs, 2006). The

constraint nudges predictions to vary with temperature and humidity in the CC-consistent *direction* (qualitative coupling), rather than enforcing a strictly units-exact magnitude. By encoding this as a soft constraint, our approach enforces physically consistent humidity–temperature relationships without restricting the network architecture or requiring explicit PDE supervision.

Using ERA5 pressure-level data (Hersbach et al., 2020), we evaluate two protocols: (i) *Global in-distribution* (random global split), and (ii) *Polar→Tropics OOD*, where models train and validate on polar latitudes only ($|\phi| \geq 66.33°$) and are tested in the tropics ($|\phi| \leq 23.5°$); midlatitudes are reported as a secondary OOD band (Stocker et al., 2013). We report *area-weighted* RMSE with cosine-latitude weights, a standard metric for model evaluation (Gleckler et al., 2008), and aggregate results over twenty five random seeds, i.e., different random initialisations of network weights and shuffles of training data. ERA5 is a reanalysis product, but the cloud-fraction target we use is not a direct observation: it is a diagnostic field produced by the prognostic cloud scheme in the ECMWF Integrated Forecasting System (Tiedtke, 1993) during data assimilation. In this sense, our model is effectively emulating that operational cloud parameterisation as constrained by observations. Our contributions are:

- A minimal-intrusion physics regulariser based on the CC relation, applicable to any feed-forward architecture without modifying the forward pass.

- A systematic OOD benchmark for cloud fraction prediction using ERA5 reanalysis data, in which all tropical and mid-latitude data (-66.33° to 66.33°) is excluded from training to induce a strong covariate shift.

- Compared with a capacity-matched MLP (i.e., a network with the same architecture and number of parameters as CC-PINN), CC-PINN preserves global in-distribution parity (it performs equally well on test data drawn from the training spatial distribution) and reduces area-weighted RMSE by ~12.6% globally and ~23.6% in the tropics under Polar→Tropics OOD.

- Ablation studies demonstrating that the OOD gains are no longer present when the physics term is removed, highlighting the role of the inductive bias (i.e., a built-in modeling assumption guiding the learning). This indicates that improvement stems from the CC-guided bias rather than capacity or sampling artefacts.

By framing cloud fraction prediction as a case study in physics-guided OOD generalisation, our work contributes to the broader ML discourse on targeted inductive biases. These results indicate that small, well-chosen physics constraints can materially improve generalisation in complex, data-limited scientific tasks, with implications for other domains facing similar challenges.

## 2 RELATED WORK

**Physics-informed neural networks.**  Physics-informed neural networks (PINNs) embed physical knowledge during training, classically by penalising residuals of governing equations (Raissi et al., 2019). Alongside full PDE-residual supervision, lighter-weight constraints have been used to stabilise learning and improve robustness in scientific settings without modifying model architecture (Beucler et al., 2021; Karpatne et al., 2017). We follow this latter line: our method adds a *Clausius–Clapeyron (CC)-guided* gradient constraint to a capacity-matched MLP, providing a soft inductive bias rather than enforcing a units-exact equality.

**Machine-learning components in climate models.**  A growing literature replaces or augments parameterisations in GCMs/ESMs with ML surrogates (Rasp et al., 2018; Brenowitz & Bretherton, 2019; Yuval & O'Gorman, 2020). These studies report strong in-distribution skill for subgrid processes (e.g., moist convection, boundary-layer turbulence), yet robustness can degrade under regime shifts or altered forcings (Dueben & Bauer, 2018; Beucler et al., 2024). This motivates physics-guided inductive biases that help retain fidelity when transferring across climatic regimes.

**Cloud-fraction modelling.**  Cloud fraction is a long-standing source of uncertainty in climate projections (Stephens, 2005). Cloud prediction in climate models is critical not only for estimating cloud cover itself, but also because clouds have strong influences on climate through shortwave

(SW) and longwave (LW) radiation feedbacks (Ramanathan et al., 1989; Bony et al., 2015a; Forster et al., 2021). Classical schemes in GCMs (Sundqvist et al., 1989; Smith, 1990b) are physically motivated but require empirical tuning and can exhibit regime-dependent biases, notably in the tropics and marine stratocumulus (Hourdin et al., 2017; Nam et al., 2012). Data-driven surrogates that learn cloud fraction from reanalysis or high-resolution simulations (Krasnopolsky et al., 2013; Yuval & O'Gorman, 2021) reduce heuristic assumptions but, without explicit thermodynamic constraints, may reproduce non-physical behaviour or overfit dominant correlations in the training distribution, (Dueben & Bauer, 2018).

**OOD under climate change: parallels to our set-up.**    Assessing whether a learned cloud scheme will remain reliable under warming typically requires testing out of the training distribution. In climate terms, forced warming shifts the joint distribution of temperature and humidity, with CC-implied increases in saturation vapour pressure and moisture availability (Held & Soden, 2006). Our Polar→Tropics protocol is not a future-scenario emulator, but it is a purposeful analogue: by training only on polar latitudes and evaluating in the tropics, we expose the model to thermodynamic states (higher $T$, higher $q$, different $RH$ structure) that lie outside the training envelope - precisely the kind of regime shift that challenges cloud schemes in warmer climates (Bony et al., 2015b). This thermodynamics-first OOD test isolates the humidity–temperature coupling that CC highlights, while holding other confounders fixed. As such, it complements hybrid/online evaluations and provides a controlled proxy for climate-change robustness (Koh et al., 2021).

**Clausius–Clapeyron relation and cloud formation.**    The Clausius–Clapeyron (CC) relation links temperature to the saturation vapour pressure of water, and hence to the saturation specific humidity $q_s(T, p)$. At fixed pressure it implies that the saturation specific humidity $q_s$ increases approximately exponentially with $T$, by about 6–7% per kelvin over typical tropospheric temperatures. Because cloud formation depends on the competition between the actual specific humidity $q$ and the saturation value $q_s(T, p)$, the CC relation provides a first-order constraint on how cloud fraction should respond to changes in temperature and humidity. In this work we use the CC slope to define a target temperature sensitivity for the predicted cloud fraction, and encourage the network's gradient $\partial \hat{c}/\partial T$ to align with this physically motivated dependence.

**Thermodynamic constraints and positioning.**    Recent work incorporates conservation and thermodynamic relationships to encourage physical consistency in atmospheric ML (Beucler et al., 2021; Yuval & O'Gorman, 2021). To our knowledge, however, the CC relation has not been used as a *differentiable, CC-guided gradient constraint* specifically for cloud-fraction emulation. Our contribution is to introduce such a minimal-intrusion constraint—leaving architecture and capacity unchanged—and to evaluate robustness under a challenging regime transfer. To avoid common evaluation artefacts on latitude–longitude grids, details on non-overlapping (leakage-resistant) splits and area-weighted metrics are given with the experimental protocol.

## 3    METHOD

### 3.1    PROBLEM FORMULATION

Let $x_i = (T_i, RH_i, q_i, \omega_i, p_i) \in \mathbb{R}^5$ denote ERA5 predictors (outlined in Table 1) at sample $i$ and let $c_i \in [0, 1]$ be cloud fraction on pressure levels (Hersbach et al., 2020). We learn $f_\theta : \mathbb{R}^5 \to [0, 1]$ with prediction $\hat{c}_i = f_\theta(x_i)$. Our design goals are: (i) *simplicity and fairness* via a capacity-matched MLP baseline. Fairness meaning the only systematic difference between models is the cc-slope regulariser; all other factors constant or symetrically tuned. And (ii) *robustness under thermodynamic regime shift*. The dataset and metrics are detailed in section 4.

### 3.2    MODELS

**Baseline MLP.**    A fully connected network with three hidden layers of 11 ReLU units and a sigmoid output producing $\hat{c} \in [0, 1]$. We chose 3×11 architecture via preliminary hyper-parameter tuning; other sizes like 10 or 12 neurons yielded similar validation performance. Full details of the MLP architecture and training in Appendix Tables A1, A2.

**CC-PINN.** Identical architecture; the only change is an additional *gradient supervision* term that aligns the model's temperature sensitivity with the Clausius–Clapeyron (CC) slope.

### 3.3 CC-SLOPE MATCHING (GRADIENT SUPERVISION)

We use the standard CC relation (Wallace & Hobbs, 2006) for saturation vapour pressure $e_s(T)$:

$$\frac{de_s}{dT} \;=\; \frac{L_v\, e_s}{R_v\, T^2}, \tag{1}$$

where $L_v$ is the latent heat of vaporisation, $T$ is temperature, and $R_v$ the gas constant for water vapour. This closed-form expression depends only on local thermodynamic variables; it is not obtained by solving a prognostic PDE. We define a per-sample residual that matches the network's partial derivative with respect to temperature (holding the other inputs fixed) to the CC slope:

$$r \;=\; \frac{\partial \hat{c}}{\partial T}\bigg|_{RH,\, q,\, \omega,\, p} \;-\; \frac{de_s}{dT}, \qquad L_{\text{phys}} \;=\; r^{\,2}. \tag{2}$$

We compute $\big(\partial\hat{c}/\partial T\big)\big|_{RH,q,\omega,p}$ via automatic differentiation with $RH$, $q$, $\omega$, and $p$ treated as constants at their sample values (i.e., no graph path from $T$ into those tensors). This yields the intended *partial* derivative. The quantity $L_{\text{phys}}$ in Eq. (2) is therefore not a PDE residual in the classical PINN sense, but a gradient-alignment term that nudges $\partial\hat{c}/\partial T$ toward the Clausius–Clapeyron slope.

Although $de_s/dT$ (Pa K$^{-1}$) and $\partial\hat{c}/\partial T$ (K$^{-1}$) carry different units, this only sets the overall scale of the penalty: any fixed unit conversion (e.g., dividing $de_s/dT$ by a reference pressure $e_0$) is a constant rescaling that $\alpha$ (a tunable weight controlling the strength of the physics term) absorbs. If $s \equiv de_s/dT$ and we replace $s$ by $\kappa s$ for any fixed $\kappa > 0$, the term becomes $\alpha\,(g - \kappa s)^2 = \alpha\,\kappa^2\,(g/\kappa - s)^2$ with $g \equiv \partial\hat{c}/\partial T$. Thus the absolute scale of the CC slope only sets the effective weight of the constraint; tuning $\alpha$ compensates for $\kappa$.[1]

As inputs are min–max normalised, $X' = (X - X_{\min})/(X_{\max} - X_{\min})$, we convert network Jacobians to physical units before equation 2.

### 3.4 TRAINING OBJECTIVE

We minimise a normalised, area-weighted objective over minibatch $\mathcal{B}$:

$$L \;=\; \frac{1}{\sum_{i\in\mathcal{B}} w_i} \sum_{i\in\mathcal{B}} w_i\Big((c_i - \hat{c}_i)^2 \;+\; \alpha\, L_{\text{phys},i}\Big), \qquad w_i = \cos\phi_i, \tag{3}$$

where $\phi_i$ is latitude (in radians). Using the same weighting at training and evaluation so the optimisation objective matches the evaluation objective, avoiding train–eval metric mismatch ('metric drift').

### 3.5 OPTIMISATION

We use Adam (Kingma & Ba, 2015) with early stopping on validation error. The learning rate, physics weight $\alpha$, dropout rate, and batch size are tuned by Bayesian optimisation (Akiba et al., 2019) on the train-validation split, see Appendix Table A2. All other hyperparameters are fixed and shared across models, seen in Appendix Table A1. We run twenty five seeds and report mean±standard error of the mean (SEM).

## 4 DATA AND EXPERIMENTAL PROTOCOL

### 4.1 ERA5 VARIABLES AND UNITS

We use ERA5 reanalysis on pressure levels at $0.25° \times 0.25°$ resolution. Inputs are air temperature $T$ (K), relative humidity $RH$ (%), specific humidity $q$ (kg kg$^{-1}$), pressure vertical velocity $\omega$ (Pa s$^{-1}$), and the pressure level index $p$ (hPa). The target $c \in [0,1]$ is ERA5 cloud fraction on pressure levels.

---

[1]This argument is at the loss level; optimiser details can change step sizes, but cross-validated $\alpha$ reliably absorbs constant rescalings of $s$.

Pressure levels used in this work: 1000, 975, 950, 925, 900, 875, 850, 825, 800, 775, 750, 700, 650, 600, 550, 500, 450, 400, 350, 300, 250, 225, 200, 175, 150, 125, 100, 70, 50, 30, 20, 10, 7, 5, 3, 2, 1 hPa.

Table 1: ERA5 predictors and target used in this study. Features are min–max normalised using *training-split only* statistics; target remains dimensionless (fraction).

| Symbol | ERA5 short name | Unit | Level | Notes |
|--------|-----------------|------|-------|-------|
| $T$ | `t` | K | pressure | Air temperature |
| $RH$ | `r` | % | pressure | Relative humidity (0–100) |
| $q$ | `q` | $\mathrm{kg\,kg^{-1}}$ | pressure | Specific humidity |
| $\omega$ | `w` | $\mathrm{Pa\,s^{-1}}$ | pressure | Pressure vertical velocity |
| $p$ | `level` | hPa | pressure | Pressure level identifier |
| $c$ (target) | `cc` | [0,1] | pressure | Cloud fraction (fraction) |

Data preprocessing is described in appendix equation B1.

## 4.2 Temporal set-up

To enforce temporal disjointness between training and evaluation, we use a single timestamp for training/validation:



Train/val: **2024-08-01 14:00** UTC.



For evaluation, we use 160 additional hourly ERA5 timesteps spanning 1950–2022, sampled across different times, months, and seasons. None of these evaluation timestamps are used during training. This design retains a strong northern hemisphere boreal-summer training snapshot while testing robustness across a much broader range of diurnal and seasonal conditions. Details for evaluation timestamps can be found in appendix B1.

## 4.3 Latitude bands and regions of assessment

We report both *global* and *band-wise* scores to expose regime dependence. Unless stated otherwise, band edges are the canonical



**Tropics:** $|\phi| \le 23.5°$,    **Midlatitude:** $23.5° < |\phi| < 66.33°$,    **Polar:** $|\phi| \ge 66.33°$,



with $\phi$ the geographic latitude. Band-wise metrics restrict the evaluation set $\mathcal{S}$ to the band and *renormalise* cosine weights within that band (Eq. 4), yielding $\mathrm{RMSE}_w^{(\mathrm{band})}$. This breakdown highlights thermodynamic contrasts (e.g., higher $T/q$ structure in the Tropics) and complements the global score.

## 4.4 Sampling and splits (leakage-robust)

Each sample is a unique spatio–temporal–pressure coordinate $(\phi, \lambda, p, t)$ (latitude, longitude, pressure level, time). Exact duplicate coordinates are removed prior to splitting. We split *by coordinates* so no $(\phi, \lambda, p, t)$ appears in more than one split:

- Train/validation (temporal slice): all coordinates with $t = 2024$-08-01 14:00 are grouped and partitioned 80/20 for train/val.

- Test (temporal hold-out): all coordinates from 160 additional ERA5 timesteps, sampled across multiple decades and seasons, form the test set and are never used for tuning, early stopping, or normaliser fitting.

When sub-sampling for efficiency, we draw samples without replacement *within each timestep*, so that the temporal split between training and test data is strictly preserved. For band-wise reporting, we then aggregate test errors over latitude bands (Tropics, Midlatitudes, Polar) using only the held-out test timesteps; the temporal split itself is not altered.

### 4.5 Model selection and reporting

Model selection (early stopping; Bayesian optimisation of learning rate and $\alpha$) is performed on the train-validation split from 2024-08-01 14:00 only. All results are reported as mean±SEM over twenty five random seeds.

### 4.6 Metrics

Because a latitude–longitude grid over-represents high latitudes, we report *area-weighted* RMSE with cosine-latitude weights. Let $\mathcal{B}$ denote the evaluation set (global or a latitude band) and $\phi_i$ the latitude of sample $i$ in radians. Define weights:

Define weights:

$$w_i = \cos(\phi_i), \tag{4}$$

and area-weighted RMSE:

$$\text{RMSE}_w = \sqrt{\frac{\sum_{i \in B} w_i (c_i - \hat{c}_i)^2}{\sum_{i \in B} w_i}}. \tag{5}$$

We use the same weighting scheme in the training loss to align optimisation and evaluation.

## 5 Results

We evaluate two protocols: (i) **Global train**, where a single timestep (2024-08-01 14:00 UTC) is used for training/validation, and (ii) **Polar train**, a spatial OOD variant where training/validation is restricted to $|\phi| \geq 66.33°$ on the same timestep. In both cases, generalisation is assessed on 160 temporally disjoint ERA5 timesteps sampled across multiple decades and seasons. Metrics are *area-weighted* RMSE, bias, and spatial correlation with cosine-latitude weights; unless stated otherwise, we report averages over the 160 evaluation timesteps (mean±SEM over 25 seeds).

In Fig.1, we report area-weighted RMSE (lower is better) by **test region** for each **model/training regime**. This presentation makes both central tendency and variability across random initialisations/data shuffles explicit, and allows direct comparison of in-distribution and out-of-distribution evaluations.

### 5.1 Global train (temporal split): parity with the baseline

Over the test timestamps, the global trained CC-PINN and the baseline are statistically comparable. Globally, CC-PINN achieves a predictive error of 0.1000±0.0002 vs 0.1028±0.0.0006 for the baseline. Band-wise: Tropics 0.0853±0.0002 vs 0.0866±0.0005, Midlatitude 0.0980±0.0002 vs 0.1006±0.0006, and Polar 0.1486±0.0003 vs 0.1556±0.0014 ). Overall, the CC term maintains in-distribution parity, and demonstrates minor improvements in accuracy and spread under the temporal shift.

### 5.2 Polar train (spatial OOD): CC-PINN improves transfer

When trained only on Polar latitudes and evaluated globally, CC-PINN improves the global score from 0.1286±0.0018 to 0.1115±0.0008 (∼**12.3%** relative reduction). The gains concentrate where the thermodynamic shift is largest:

- **Tropics:** 0.1373±0.0034 → 0.1049±0.0015 (∼**21.4%** lower RMSE).
- **Midlatitude:** 0.1154±0.0012 → 0.1064±0.0005 (∼**7.8%** lower).
- **Polar:** a modest degradation 0.1501±0.0007 → 0.1513±0.0006 (∼**0.8%** higher), consistent with a trade-off that prioritises correct temperature sensitivity in warmer/moister regimes.

Full band-wise RMSE values are given in appendix B, table B2. These results indicate that *aligning $\partial \hat{c}/\partial T$ with the CC slope* meaningfully improves extrapolation to thermodynamically distinct states.

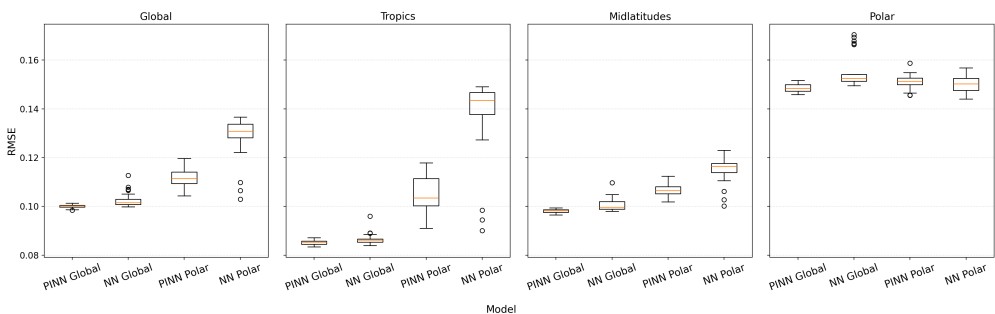

Figure 1: Area-weighted RMSE (lower is better) by **test region** for each **model/training regime**. Small circles are seed-wise results ($n = 25$). Boxes show the median (line) and interquartile range (box); whiskers use $1.5 \times \text{IQR}$.

### 5.3 SIGNIFICANCE OF IMPROVEMENTS

Polar–train global reduction ($0.1275 \rightarrow 0.1119$) is significant by a two-sample Welch t-test, $t(32.62) = -8.17$ (two-sided $p = 2.17 \times 10^{-9}$); the PINN mean RMSE is lower than the NN, with a large effect (Cohen's $d \approx 2.31$).

### 5.4 GRADIENT ALIGNMENT WITH CC

We verify that gradient supervision changes the model's temperature sensitivity in the intended direction. Define a tolerance-aware sign function

$$\text{sign}_\tau(x) = \begin{cases} -1, & x \leq -\tau, \\ 0, & |x| < \tau, \\ 1, & x \geq \tau, \end{cases} \tag{6}$$

where $\tau > 0$ is a small tolerance chosen so that near-zero gradients are treated as agreement rather than penalised for arbitrary sign.

Let $g_i = \frac{\partial \hat{c}}{\partial T}\big|_i$ and $s_i = \frac{de_s}{dT}\big|_i$. We report the directional agreement (area-weighted):

$$\text{DA}_w = \frac{\sum_{i \in \mathcal{S}} w_i \, \mathbf{1}\big[\text{sign}_{\tau_g}(g_i) = \text{sign}_{\tau_s}(s_i)\big]}{\sum_{i \in \mathcal{S}} w_i}, \tag{7}$$

with $w_i = \cos \phi_i$ and $\tau_g$ and $\tau_s$ are small tolerances applied to the model gradient and CC slope, respectively.

where $\tau > 0$ is a small tolerance chosen so that near-zero gradients are treated as agreement rather than penalised for arbitrary sign.

and the Spearman correlation $\rho$ between $\partial \hat{c}/\partial T$ and $de_s/dT$ across samples. Under **Polar train**, CC-PINN improves DA and $\rho$ in the Tropics and Midlatitude (Table 2), indicating the loss steers sensitivities toward CC-consistent behaviour.

Table 2: Clausius–Clapeyron alignment under **Polar train**: area-weighted directional agreement ($\text{DA}_w$) and Spearman $\rho(g, s)$ (mean $\pm$ SEM over 25 seeds. Higher is better.

| Region | PINN $\text{DA}_w$ | PINN $\rho$ | NN $\text{DA}_w$ | NN $\rho$ |
|---|---|---|---|---|
| Tropics | $0.554 \pm 0.056$ | $0.191 \pm 0.035$ | $0.260 \pm 0.057$ | $0.034 \pm 0.043$ |
| Midlatitude | $0.647 \pm 0.053$ | $0.135 \pm 0.038$ | $0.385 \pm 0.055$ | $-0.008 \pm 0.043$ |
| Polar | $0.613 \pm 0.050$ | $0.125 \pm 0.038$ | $0.405 \pm 0.050$ | $-0.017 \pm 0.039$ |
| Global | $0.613 \pm 0.051$ | $0.154 \pm 0.035$ | $0.359 \pm 0.053$ | $0.013 \pm 0.042$ |

Table 3: Temperature–stratified errors using **equal-weight** bins (bin centres shown, in K). RMSE is computed within each bin; $\Delta\% = 100\,(\text{RMSE}_{\text{NN}} - \text{RMSE}_{\text{PINN}})/\text{RMSE}_{\text{NN}}$ (higher is better).

| Bin centre (K) | CC-PINN RMSE | NN RMSE | $\Delta\%$ |
|---|---|---|---|
| 199.49 | 0.1115 | 0.1435 | 22.3 |
| 224.80 | 0.1029 | 0.1505 | 31.6 |
| 250.12 | 0.1189 | 0.1291 | 7.9 |
| 275.43 | 0.1188 | 0.1151 | -3.2 |
| 300.74 | 0.0998 | 0.0933 | -7.0 |

## 5.5 WHERE DO GAINS ARISE? STRATIFICATION BY TEMPERATURE

We stratify the test set into equal-area temperature bins and report area-weighted RMSE within each bin. Contrary to the simplest "warmer $\Rightarrow$ larger gains" expectation, the largest relative reductions occur in the coldest bins, while improvements are modest in the warmest bin. This pattern is consistent with the dual role of our CC-slope term: (i) in cold regimes where $de_s/dT \approx 0$, it damps spurious temperature sensitivity by nudging $\partial\hat{c}/\partial T \to 0$, yielding sizable error reductions; (ii) in warm regimes — often dry, the baseline already attains low RMSE and cloud fraction is more dynamics-limited than thermodynamically limited, so aligning the temperature sensitivity offers limited additional benefit. Importantly, the warmest bin's *absolute* errors are already small, so relative gains naturally appear muted even when absolute gaps are comparable across bins. Overall, the stratified view reinforces that CC-slope supervision improves robustness where $T$-sensitivity is most error-prone (cold/mid bins) while preserving parity in warm states. Full results are shown in Table 3.

Maps showing the spatial distribution of RMSE, Correlation, and Bias can be seen in appendix B B1, B2, B3.

Removing the CC-slope term ($\alpha{=}0$) erases OOD gains (Polar trained Tropics and Midlatitude), confirming that improvements derive from the gradient supervision rather than capacity or sampling.

## 5.6 TRAINING STABILITY

CC-PINN reduces variance across seeds in the OOD setting (e.g., Tropics SEM 0.0015 vs 0.0034; Global SEM 0.0008 vs 0.0018 under Polar train), suggesting the CC term regularises optimisation by discouraging spurious temperature responses. Full band-wise RMSE values are given in appendix B, table B2.

# 6 DISCUSSION

**Key finding.** A single, lightweight CC–slope supervision (Eq. 2) materially improves extrapolation to thermodynamically distinct states. Under **Polar train**, CC-PINN reduces *global* RMSE from 0.1286±0.0018 to 0.1115±0.0008 ($\sim$12.6%), with the largest gain in the *Tropics* (0.1373±0.0034 $\to$ 0.1049±0.0015; $\sim$23.6%). Under **Global train** the two models are statistically comparable, indicating the constraint does not harm in-distribution performance.

**Warm-bin trade-off.** Our temperature-stratified results show the largest gains in cold bins and near parity in the warmest bin, which aligns with CC-slope supervision acting chiefly as a guardrail against spurious $T$-sensitivity in regimes where $de_s/dT$ is small, and offering limited headroom where baseline errors are already low and clouds are dynamics-dominated.

## 6.1 IMPLICATIONS FOR OOD GENERALISATION

These results support the broader claim that *task-relevant* inductive biases can curb spurious correlations that otherwise limit transfer. Here, aligning $\partial\hat{c}/\partial T$ to the CC slope produces gains concentrated in warm/moist regimes—the very states where CC effects amplify humidity availability—while leaving the temporal split essentially unchanged. This specificity matters: the constraint

targets the mechanism most likely to shift under warming, rather than imposing a broad architectural prior.

## 6.2 RELEVANCE TO CLIMATE MODELLING

For climate applications, the constraint is *deployable*: it leaves the forward pass untouched, adds negligible overhead, and can be switched on/off via a single weight $\alpha$. As such, it is compatible with hybrid deployments (residual correction or emulator settings). That said, our present evaluation is *offline* and uncoupled; online stability and conservation in a prognostic model remain to be tested. A practical next step is single-column online coupling (e.g., SCM or aquaplanet), where the CC-guided sensitivity can be stress-tested under feedbacks with radiation and dynamics (Brenowitz & Bretherton, 2019; Rasp et al., 2018).

## 6.3 METHODOLOGICAL STRENGTHS

(i) *Capacity fairness:* identical depth/width across models isolates the effect of the physics term. (ii) *Leakage-robust evaluation:* grouped splits over $(\phi, \lambda, p, t)$ with train-only normalisation, and area-weighted metrics. (iii) *Two axes of shift:* a temporal axis (from a single August training timestep to 160 evaluation timesteps across decades and seasons) and a spatial OOD axis (Polar train), with parity under global training and clear gains for polar→tropics transfer. (iv) *Compute efficiency:* all runs are desktop-trainable, supporting reproducibility.

## 6.4 LIMITATIONS AND THREATS TO VALIDITY

- **Target fidelity.** ERA5 cloud fraction is model-derived; learned biases may reflect the re-analysis operator. At the same time, this means our emulator is effectively trained on the IFS cloud parameterisation used operationally at ECMWF, so our results should be interpreted as learning and regularising a state-of-the-art scheme rather than direct observations. Observation-based targets (e.g., CloudSat/Calipso or satellite retrievals (Stephens et al., 2002; Winker et al., 2010)) would strengthen external validity.

- **Temporal coverage.** We deliberately train on a single polar timestep, and evaluate on 160 timesteps across multiple decades and seasons. This isolates the effect of the CC-based constraint but does not yet explore multi-year or multi-decadal training, which is an important avenue for future work.

- **Deterministic vs probabilistic prediction.** Another natural extension is to probabilistic forecasting (e.g., quantile or distributional prediction of cloud fraction). In this work we deliberately fix a deterministic setup to isolate the effect of the CC-based gradient constraint; incorporating uncertainty while preserving physically guided temperature sensitivity is an important direction for future work.

Our use of a small ($< 500$-parameter) MLP is intentional: it reflects the computational constraints of hybrid climate models and isolates the effect of the CC-based loss from architectural changes. The CC constraint itself is architecture-agnostic and could, in principle, be applied to larger networks; evaluating this scaling is an important avenue for future work.

## 6.5 BROADER IMPACT

Physics-guided losses like the CC term provide a simple bridge between domain knowledge and ML robustness. Because they are transparent, tunable, and architecture-agnostic, they are accessible to groups without large compute budgets and can foster more trustworthy ML components in climate workflows. Care is still required: improvements here do not guarantee safe behaviour in coupled models or extremes; rigorous online testing and calibration are prerequisites for operational use.

**Take-away.** A minimal, interpretable constraint on temperature sensitivity—implemented as gradient supervision—delivers measurable OOD gains where they matter most, without architectural changes. This pattern (soft, mechanism-targeted physics guidance) is a pragmatic path for advancing robust scientific ML.

# 7 CONCLUSION

We introduced CC-PINN, which embeds the Clausius–Clapeyron (CC) law as a gradient-based supervision term on temperature sensitivity. With no architectural changes and negligible overhead, this *CC-slope* constraint preserves parity under a temporal split while improving extrapolation under a spatial OOD stress test: under **Polar train**, global RMSE drops from $0.1286\pm0.0018$ to $0.1115\pm0.0008$ ($\sim12.6\%$), with the largest gain in the **Tropics** ($0.1373\pm0.0034 \rightarrow 0.1049\pm0.0015$; $\sim23.6\%$). A modest Polar-band trade-off is observed, consistent with prioritising correct warm-state sensitivity.

**Takeaways.** (1) A small, task-relevant physics prior can cut OOD error without degrading in-distribution performance, with benefits concentrated in regimes where CC effects are strongest. (2) Gains are robust across seeds and capacity controls; they stem from the gradient supervision rather than model size or sampling, and are insensitive to constant rescalings of the CC target (absorbed by $\alpha$). (3) The approach is lightweight, architecture-agnostic, and compute-efficient, making it practical for broader scientific ML use. Overall, the focus of this work is on how a simple, interpretable physics regulariser changes generalisation behaviour for a fixed architecture, rather than on competing with state-of-the-art cloud schemes.

**Future work.** Extend to multi-constraint objectives (e.g., simple moisture/energy closures or monotonicity in $RH$), evaluate against observation-based cloud products across more times/years and report calibration, and test online in single-column/GCM settings to assess stability, conservation, and long-horizon forecast skill.

REPRODUCIBILITY STATEMENT.

We specify the dataset, variables, and units in Sec. 4.1; exact timestamps and latitude-band definitions in Secs. 4.2–4.3; preprocessing and normalisation in Sec. 4.4; leakage-robust sampling/splits and model-selection protocol in Secs. 4.5–4.6; and the area-weighted RMSE metric (cosine-latitude weights) in Sec. 4.7 with formulas in Eqs. (5)–(6). Model architectures, losses, and optimisation are detailed in Sec. 3 (CC slope Eq. (1), gradient supervision Eq. (2), training objective Eq. (3); optimiser/early stopping in Secs. 3.4–3.5), with fixed settings and tuned hyperparameters in Appendix Tables A1–A2. We report mean ± SEM over 25 seeds, use a temporal hold-out test set unseen by tuning, and provide significance tests (Sec. 5.3), gradient-alignment diagnostics (Sec. 5.4), temperature-stratified analyses (Sec. 5.5), and stability notes (Sec. 5.6). ERA5 acquisition is reproducible via the Copernicus Climate Data Store using the variable names, levels, and exact times listed in Secs. 4.1–4.2.

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

# A   APPENDIX

DECLARATION OF GENERATIVE AI AND AI-ASSISTED TECHNOLOGIES IN THE WRITING PROCESS.

We used ChatGPT (GPT-4 and GPT-5) in a limited editorial role: (1) code refactoring suggestions for non-novel boilerplate (renaming variables, reformatting functions, extracting helpers); (2) light text polishing for grammar/clarity; and (3) LaTeX formatting guidance (environments, floats, and bibliography style). We did *not* use AI tools to propose or validate methods, derive results, tune hyperparameters, generate figures/tables, create datasets/labels, or perform literature reviews without manual verification. All AI-assisted changes were inspected and, where appropriate, rewritten by the authors; the authors accept full responsibility for the content.

Table A1: Fixed training settings shared by all runs.

| Item | Value |
|---|---|
| Architecture | 3×11 ReLU; sigmoid output |
| Optimiser | Adam |
| Epochs (max) | 1000 (early stopping; patience 20 epochs) |
| Seeds | 25 (report mean±SEM) |
| Normalisation | Per-feature min–max (train split only; frozen for val/test) |
| Area weighting | $w_i = \cos(\phi_i)$ (radians) |

Table A2: Tuned hyperparameters by model and training protocol. Values selected on validation and then fixed for 25 seeds; all other settings are shared (see Table A1).

| Hyperparameter | Global train | | Polar train | |
|---|---|---|---|---|
| | **PINN** | **NN** | **PINN** | **NN** |
| Learning rate $\eta$ | $1.461 \times 10^{-3}$ | $5.285 \times 10^{-3}$ | $5.659 \times 10^{-4}$ | $1.858 \times 10^{-3}$ |
| Batch size $B$ | 16 | 64 | 32 | 16 |
| Physics weight $\alpha$ | $1.015 \times 10^{-4}$ | — | $1.010 \times 10^{-4}$ | — |
| Dropout rate | 0.0112 | 0.0118 | 0.00770 | 0.0206 |

# B   APPENDIX

## B.1   PREPROCESSING AND NORMALISATION

Per-feature min–max scalers are fit on the *training split only* and then frozen:

$$X' = \frac{X - X_{\min}}{X_{\max} - X_{\min}}, \tag{B1}$$

where $(X_{\min}, X_{\max})$ are computed on training data from 2024-08-01 14:00. We pass $RH$ internally as a fraction in $[0, 1]$ (not %) for consistent gradient units.

Table B1: Timestamps used in the test dataset. IDs are encoded as YYYYMMDD_hHH.

| # | ID | # | ID | # | ID | # | ID |
|---|---|---|---|---|---|---|---|
| 1 | 19500413_h23 | 41 | 19700707_h21 | 81 | 19900503_h06 | 121 | 20101119_h02 |
| 2 | 19500511_h00 | 42 | 19701123_h07 | 82 | 19900511_h02 | 122 | 20110610_h21 |
| 3 | 19510119_h06 | 43 | 19710224_h23 | 83 | 19900918_h12 | 123 | 20111108_h08 |
| 4 | 19510402_h23 | 44 | 19711029_h11 | 84 | 19910415_h21 | 124 | 20120325_h20 |
| 5 | 19520703_h04 | 45 | 19711101_h14 | 85 | 19910721_h16 | 125 | 20120907_h01 |
| 6 | 19520810_h16 | 46 | 19711202_h17 | 86 | 19910919_h20 | 126 | 20120913_h18 |
| 7 | 19530131_h07 | 47 | 19720722_h09 | 87 | 19911018_h21 | 127 | 20140623_h21 |
| 8 | 19530213_h00 | 48 | 19731226_h06 | 88 | 19921026_h17 | 128 | 20140930_h06 |
| 9 | 19540915_h07 | 49 | 19740404_h08 | 89 | 19940122_h07 | 129 | 20141130_h16 |
| 10 | 19550114_h18 | 50 | 19740406_h08 | 90 | 19940413_h19 | 130 | 20150501_h17 |
| 11 | 19560212_h02 | 51 | 19750201_h20 | 91 | 19940926_h19 | 131 | 20151202_h19 |
| 12 | 19560417_h06 | 52 | 19761031_h20 | 92 | 19950402_h16 | 132 | 20160118_h23 |
| 13 | 19560815_h13 | 53 | 19770306_h22 | 93 | 19950715_h02 | 133 | 20160417_h22 |
| 14 | 19561001_h00 | 54 | 19770408_h02 | 94 | 19970818_h23 | 134 | 20170415_h11 |
| 15 | 19570304_h00 | 55 | 19770708_h08 | 95 | 19980603_h08 | 135 | 20170926_h06 |
| 16 | 19570304_h03 | 56 | 19771114_h21 | 96 | 19980623_h01 | 136 | 20171230_h09 |
| 17 | 19570803_h23 | 57 | 19780306_h07 | 97 | 19980815_h20 | 137 | 20180317_h15 |
| 18 | 19571112_h17 | 58 | 19780405_h21 | 98 | 19990827_h00 | 138 | 20180713_h17 |
| 19 | 19580111_h20 | 59 | 19790510_h11 | 99 | 19990829_h03 | 139 | 20180803_h04 |
| 20 | 19580405_h03 | 60 | 19790920_h03 | 100 | 19991204_h17 | 140 | 20190225_h02 |
| 21 | 19610222_h02 | 61 | 19800928_h06 | 101 | 20000103_h19 | 141 | 20200327_h18 |
| 22 | 19610928_h06 | 62 | 19801222_h13 | 102 | 20000114_h23 | 142 | 20200829_h07 |
| 23 | 19620930_h04 | 63 | 19820727_h01 | 103 | 20000320_h03 | 143 | 20200919_h10 |
| 24 | 19630131_h07 | 64 | 19821007_h23 | 104 | 20011009_h14 | 144 | 20201003_h01 |
| 25 | 19631025_h08 | 65 | 19821212_h23 | 105 | 20020325_h04 | 145 | 20201017_h22 |
| 26 | 19640110_h11 | 66 | 19821220_h04 | 106 | 20020810_h19 | 146 | 20210201_h21 |
| 27 | 19640330_h02 | 67 | 19830712_h06 | 107 | 20021214_h16 | 147 | 20210625_h23 |
| 28 | 19640405_h03 | 68 | 19840608_h20 | 108 | 20030417_h13 | 148 | 20220218_h03 |
| 29 | 19660105_h03 | 69 | 19840701_h08 | 109 | 20030507_h20 | 149 | 20220621_h00 |
| 30 | 19660311_h09 | 70 | 19850222_h04 | 110 | 20030820_h15 | 150 | 20220621_h18 |
| 31 | 19660622_h06 | 71 | 19860211_h18 | 111 | 20031025_h03 | 151 | 20220731_h18 |
| 32 | 19661008_h08 | 72 | 19860331_h07 | 112 | 20040126_h09 | 152 | 20220901_h08 |
| 33 | 19661208_h11 | 73 | 19860418_h17 | 113 | 20040311_h05 | 153 | 20221012_h07 |
| 34 | 19671030_h13 | 74 | 19860512_h22 | 114 | 20050831_h00 | 154 | 20230915_h02 |
| 35 | 19671125_h02 | 75 | 19860718_h12 | 115 | 20050910_h03 | 155 | 20240914_h14 |
| 36 | 19680307_h14 | 76 | 19860722_h13 | 116 | 20050910_h19 | 156 | 20241031_h11 |
| 37 | 19680705_h05 | 77 | 19870508_h15 | 117 | 20060118_h16 | 157 | 20250123_h03 |
| 38 | 19680724_h10 | 78 | 19870905_h10 | 118 | 20080717_h05 | 158 | 20250420_h07 |
| 39 | 19690118_h01 | 79 | 19890110_h10 | 119 | 20091005_h20 | 159 | 20250421_h13 |
| 40 | 19690420_h20 | 80 | 19890614_h01 | 120 | 20091027_h23 | 160 | 20250611_h06 |

Table B2: Area-weighted RMSE (mean $\pm$ SEM over 25 seeds.). Bands: Tropics $|\phi| \leq 23.5°$, Midlatitude $23.5° < |\phi| < 66.33°$, Polar $|\phi| \geq 66.33°$. Lower is better.

| Region | Global train | | Polar train | |
| | PINN | NN | PINN | NN |
|---|---|---|---|---|
| Tropics | $0.0853 \pm 0.0002$ | $0.0866 \pm 0.0005$ | $0.1049 \pm 0.0015$ | $0.1373 \pm 0.0034$ |
| Midlatitude | $0.0980 \pm 0.0002$ | $0.1006 \pm 0.0006$ | $0.1064 \pm 0.0005$ | $0.1154 \pm 0.0012$ |
| Polar | $0.1486 \pm 0.0003$ | $0.1556 \pm 0.0014$ | $0.1513 \pm 0.0006$ | $0.1501 \pm 0.0007$ |
| Global | $0.1000 \pm 0.0002$ | $0.1028 \pm 0.0006$ | $0.1115 \pm 0.0008$ | $0.1286 \pm 0.0018$ |

Table B3: Area-weighted bias (mean ± SEM over 25 seeds.). Bands: Tropics $|\phi| \leq 23.5°$, Midlatitude $23.5° < |\phi| < 66.33°$, Polar $|\phi| \geq 66.33°$. Closer to 0 is better.

| | Global train | | Polar train | |
| --- | --- | --- | --- | --- |
| **Region** | **PINN** | **NN** | **PINN** | **NN** |
| Tropics | $-0.0074 \pm 0.0004$ | $-0.0047 \pm 0.0005$ | $-0.0196 \pm 0.0016$ | $-0.0349 \pm 0.0018$ |
| Midlatitude | $-0.0048 \pm 0.0004$ | $-0.0035 \pm 0.0005$ | $-0.0113 \pm 0.0006$ | $-0.0165 \pm 0.0007$ |
| Polar | $0.0089 \pm 0.0009$ | $0.0105 \pm 0.0012$ | $0.0049 \pm 0.0008$ | $-0.0029 \pm 0.0010$ |
| Global | $-0.0044 \pm 0.0004$ | $-0.0025 \pm 0.0004$ | $-0.0129 \pm 0.0009$ | $-0.0224 \pm 0.0011$ |

Table B4: Area-weighted correlation $\rho$ between predictions and targets (mean ± SEM over 25 seeds.). Bands: Tropics $|\phi| \leq 23.5°$, Midlatitude $23.5° < |\phi| < 66.33°$, Polar $|\phi| \geq 66.33°$. Higher is better.

| | Global train | | Polar train | |
| --- | --- | --- | --- | --- |
| **Region** | **PINN** | **NN** | **PINN** | **NN** |
| Tropics | $0.8149 \pm 0.0010$ | $0.8069 \pm 0.0026$ | $0.7307 \pm 0.0059$ | $0.3980 \pm 0.0339$ |
| Midlatitude | $0.8625 \pm 0.0005$ | $0.8551 \pm 0.0018$ | $0.8371 \pm 0.0017$ | $0.8063 \pm 0.0040$ |
| Polar | $0.8298 \pm 0.0007$ | $0.8107 \pm 0.0037$ | $0.8232 \pm 0.0013$ | $0.8240 \pm 0.0019$ |
| Global | $0.8445 \pm 0.0005$ | $0.8347 \pm 0.0024$ | $0.8049 \pm 0.0029$ | $0.7319 \pm 0.0077$ |

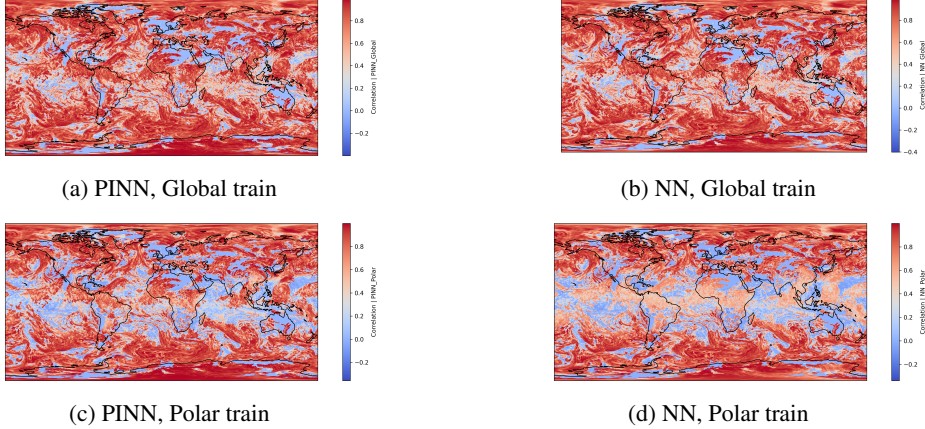

(a) PINN, Global train

(b) NN, Global train

(c) PINN, Polar train

(d) NN, Polar train

Figure B1: Global maps of mean RMSE, averaged across the evaluation set, for each model and training protocol.

(a) PINN, Global train

(b) NN, Global train

(c) PINN, Polar train

(d) NN, Polar train

Figure B2: Global maps of mean correlation between predictions and targets, averaged across the evaluation set.

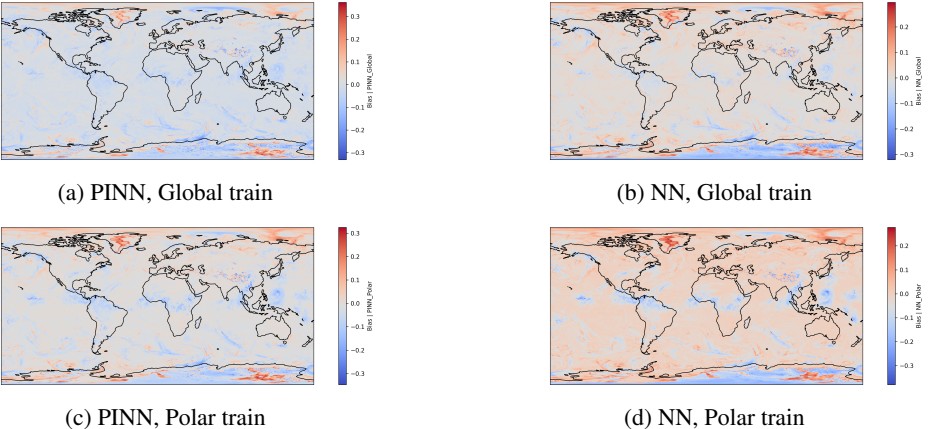

(a) PINN, Global train

(b) NN, Global train

(c) PINN, Polar train

(d) NN, Polar train

Figure B3: Global maps of mean bias (predicted minus target cloud fraction), averaged across the evaluation set.

