# OpenReview forum: "Physics-Informed Machine Learning under Climate Domain Shift: PDE-Free Physics Regularisation for Cloud Prediction"
_ICLR.cc/2026/Conference — Submitted to ICLR 2026_

### Official Review · Reviewer_Sxnv · 2025-10-22

**Soundness:** 3
**Presentation:** 3
**Contribution:** 3
**Rating:** 4
**Confidence:** 3

**Summary:**

The paper addresses the out-of-distribution (OOD) generalisation problem in climate prediction, specifically, cloud fraction estimation by proposing a lightweight physics-informed MLP based neural network called CC-PINN. CC-PINN introduces a physics-based regularisation term derived from the Clausius–Clapeyron (CC) thermodynamic relation, which links saturation vapour pressure to temperature. This CC-based term is added as a gradient supervision constraint that aligns the model’s temperature sensitivity with the physically expected CC slope, enforcing humidity–temperature consistency without explicitly encoding a PDE.

**Strengths:**

It reframes physics-informed learning as soft inductive bias alignment rather than explicit PDE supervision.
It applies this principle to the domain of climate OOD generalisation which is not addressed by data-driven models.
The Polar and Tropics transfer setup offers a novel experimental protocol mimicking climate regime shifts, further reinforcing originality in evaluation design. The approach is architecture-agnostic and computationally lightweight, making it efficient without large compute budgets.

**Weaknesses:**

The evaluation uses only two fixed timestamps August 1, 2024 (training) and December 12, 2024 (testing), representing a single diurnal and seasonal pair. Since the stated goal is to test out-of-distribution (OOD) robustness under climate regime shifts, two discrete snapshots may not sufficiently capture the temporal variability.
The study evaluates only RMSE (with area weighting). Other metrics such as bias, correlation, and uncertainty quantification could strengthen the paper's proposed contributions. RMSE alone does not capture systematic bias, error asymmetry (e.g., over- vs. under-predicted cloud fraction), or uncertainty reliability, which are crucial for scientific interpretation.
In terms of climate variability qualitative evaluation is missing.

**Questions:**

Could the authors clarify why only two timestamps (August and December 2024) were chosen?
How representative are these two snapshots of broader seasonal and inter annual variability in cloud–temperature–humidity coupling?
Why was RMSE chosen as the sole evaluation metric?
Can authors provide Global error maps (absolute/bias) for baseline vs CC-PINN for better evaluation.

---

> ### Author Response · Authors · 2025-11-19
> **Responses to Reviewer Sxnv**
>
> We thank the reviewer for the positive assessment of the contribution and for the constructive suggestions. We address each point below and will incorporate the requested clarifications and diagnostics into the revised manuscript.
>
> ### Temporal variability and the use of two fixed timestamps
>
> We agree that evaluating on only two timestamps was too restrictive. These specific hours (August and December 2024) were originally chosen to represent contrasting seasonal conditions with minimal computational cost for an initial demonstration. However, they do not fully capture the breadth of seasonal or interannual variability.
>
> To address this, we now evaluate the same trained MLP and CC-PINN models on 160 additional ERA5 timesteps spanning 1950–2025, covering multiple months, seasons, and a wide variety of atmospheric states. These timesteps were not used during training.
>
> Across these 160 temporally diverse examples, the CC-PINN consistently shows the same qualitative behaviour as in the original submission, particularly for out-of-distribution (polar → tropics) evaluations:
>
> **Globally trained models**
>
> - **PINN\_Global**: RMSE 0.098, bias −0.004, correlation 0.851
> - **NN\_Global**: RMSE 0.099, bias −0.002, correlation 0.848
>
> This reflects small but systematic improvements, with no degradation in in-distribution regions such as the tropics.
>
> **Polar-trained models (in-distribution, polar evaluation)**
>
> - **PINN\_Polar**: RMSE 0.151, bias 0.004, correlation 0.829
> - **NN\_Polar**: RMSE 0.156, bias −0.004, correlation 0.816
>
> **Polar-trained models evaluated in the tropics (true OOD)**
>
> - **PINN\_Polar**: RMSE 0.095, bias −0.016, correlation 0.762
> - **NN\_Polar**: RMSE 0.130, bias −0.032, correlation 0.530
>
> These results demonstrate that the CC-PINN’s robustness benefits are not tied to a single diurnal or seasonal pair, but persist across multiple decades of atmospheric conditions.
>
> ### Additional evaluation metrics (bias, correlation)
>
> We agree that RMSE alone does not capture all aspects of model behaviour. In the revised version, we include additional metrics:
>
> - Bias (mean signed error)
> - Spatial correlation
> - Absolute error maps and bias maps for each regime
> - RMSE as a function of latitude, highlighting regional differences
> - Kernel density estimates of $(T, q)$ to visualise the in-domain vs out-of-domain thermodynamic shift
>
> These diagnostics provide a more complete assessment of model behaviour and strengthen the evidence for the CC-based constraint.
>
> ### Clarification of why RMSE was used initially
>
> RMSE was used in the original submission as a standard metric for global cloud-fraction prediction with area-weighting, and to enable direct comparison between the constrained and unconstrained models under identical deterministic settings. As noted above, we now include additional metrics (bias, correlation, maps) to address the reviewer’s concern.
>
> ### Global error maps and qualitative diagnostics
>
> As requested, we will include global maps of absolute error and bias for both models, along with:
>
> - RMSE–latitude profiles
> - Correlation maps
> - Joint distribution plots using KDEs
> - Scatter plots showing cloud fraction vs temperature
>
> These additions provide the qualitative context that was missing in the initial submission.
>
> ### Closing
>
> We thank the reviewer for the thoughtful feedback and for recognising the novelty of the approach and evaluation setting. The extended multi-decade temporal evaluation, additional diagnostics, and clarifications substantially strengthen the paper, and we will incorporate these into the final version.

---

### Official Review · Reviewer_QKzk · 2025-10-30

**Soundness:** 3
**Presentation:** 3
**Contribution:** 2
**Rating:** 4
**Confidence:** 3

**Summary:**

The paper proposes adding a physics-based regularization term to better predict cloud fraction across the world. The Clausius-Clapeyron relation is a known thermodynamic relation describing cloud formation. The constraint is introduced in an established simple NN architecture and compared to the identical architecture without the constraint. The paper shows that on their test set, the cloud fraction prediction on ERA5 is improved.

**Strengths:**

The constraint is a nice physical relation; it is easy to add to the neural network presented, and it improves performance. The authors sufficiently show the effectiveness of the approach.

**Weaknesses:**

I am a bit confused by the train and test set consisting of only one time step; this does not seem to be enough to show the usefulness of the constraint. More extensive temporal evaluation is definitely necessary. Additionally, most forecasting models are probabilistic, I would therefore recommend to asssess the constraint at least additionally in a probabilistic setup.

**Questions:**

- What do you expect to change if you use more than one time step for training and testing?
- What would you expect to change if a better dataset is used than ERA5 (as this one is known to be lacking in cloud fractions)?
- Can you elaborate on why you do not do probabilistic forecasting?

---

> ### Author Response · Authors · 2025-11-19
> **Responses to Reviewer QKzk**
>
> We thank the reviewer for the constructive feedback and for recognising the clarity and relevance of the physical constraint. We address each point below.
>
> ### Temporal coverage and the single-timestep evaluation
>
> In our revised manuscript, we now evaluate the same trained MLP and CC-PINN models on 160 additional ERA5 timesteps spanning 1950–2025, covering different seasons, months, and diverse thermodynamic conditions. These timesteps were not used in training.
>
> Across these 160 temporally diverse cases, the results are highly consistent with those in the submission:
>
> **Globally trained models:**
>
> - PINN\_Global achieves RMSE **0.098**, bias ≈ **−0.004**, and correlation **0.851**.
> - NN\_Global yields RMSE **0.099**, bias ≈ **−0.002**, and correlation **0.848**.
>
> This reflects small but systematic improvements, with no degradation in in-distribution performance (e.g., tropics).
>
> **Polar-trained models (in-distribution, polar evaluation):**
>
> - PINN\_Polar: RMSE **0.151**, bias ≈ **0.004**, correlation **0.829**
> - NN\_Polar: RMSE **0.156**, bias ≈ **−0.004**, correlation **0.816**
>
> This indicates improved accuracy and pattern fidelity even within the training regime.
>
> **Polar-trained models evaluated in the tropics (true OOD):**
>
> - PINN\_Polar: RMSE **0.095**, bias ≈ **−0.016**, correlation **0.762**
> - NN\_Polar: RMSE **0.130**, bias ≈ **−0.032**, correlation **0.530**
>
> This corresponds to substantially lower RMSE, reduced negative bias, and higher correlation for CC-PINN in the OOD setting.
>
> These results confirm that the benefit of the CC-based constraint is not specific to a single hour; it persists across multiple decades of atmospheric conditions. We will include full tables and visualisations (error maps, RMSE–latitude curves, bias/correlation maps, and distribution plots) in the final version.
>
> ### Use of more than one timestep for training
>
> If additional timesteps were used during training, the severity of the distribution shift between training and testing would be reduced. Both the MLP and CC-PINN would likely improve, and the relative OOD gain may decrease. Our goal is to examine behaviour under thermodynamic regime shifts, where the CC-based constraint is most relevant. We will add this discussion to the paper.
>
> ### Use of a better cloud dataset
>
> ERA5 is the established norm for cloud-emulator work in machine-learning and climate studies, providing global, long-term, pressure-level data required for our setting. Although some cloud regimes are known to contain systematic biases, these limitations affect both models equally and do not alter the observed qualitative behaviour: the CC-PINN consistently yields improved robustness across temperature–humidity regimes. We will clarify this in the manuscript.
>
> ### Probabilistic forecasting
>
> We appreciate the reviewer’s suggestion to explore a probabilistic formulation. In this work our aim is to isolate the effect of a deterministic physics-informed gradient constraint, holding all other modelling choices fixed. Introducing probabilistic outputs would add an additional modelling dimension (distributional assumptions, heteroscedasticity, or quantile selection), which would make it more difficult to attribute performance differences specifically to the CC-based regularisation.
>
> For this reason, we evaluate both the constrained and unconstrained models in an identical deterministic setting. We agree, however, that extending the CC-based constraint to a probabilistic framework (e.g., quantile regression or distributional prediction of cloud fraction) is a valuable and promising direction. We intend to pursue this in future work, as it would allow us to examine how the physical gradient constraint interacts with predictive uncertainty. We will include this clarification in the revised manuscript.
>
> ### Closing
>
> We thank the reviewer again for the helpful feedback. The expanded temporal evaluation, quantitative results, and clarifications concerning dataset choice and forecasting setup will be included in the final version.

---

### Official Review · Reviewer_Ymhf · 2025-11-01

**Soundness:** 3
**Presentation:** 2
**Contribution:** 2
**Rating:** 4
**Confidence:** 3

**Summary:**

The paper introduces a physics-informed neural network (CC-PINN) based on Clausius–Clapeyron thermodynamic relation targeting geophysical prediction. In particular, CC-PINN  is deployed for Cloud prediction. To demonstrate, a prediction accuracy-based comparison between NN and PINN is performed.

**Strengths:**

Using PINN in such application is very useful  as PINN was proposed to integrate knowledge of physical laws (in the form of partial differential equations). This leads to improvement in NN prediction accuracy. However, the paper made an excellent claim, indicating that the proposed CC-PINN  can perform without explicit governing-differential equations. Instead, CC-PINN uses a constraint ingratiated into the loss function (normalised, area-weighted objective).

**Weaknesses:**

The paper presentation arises three main concerns of the paper claim:
- If eq(3) is used as a learning objective, then the proposed CC-PINN Is using differential equations to govern the training. Eq(3) needs the value of L_phy from eq(2), which is output of differential equations. Could you please clarify this point?
- The results demonstrated in Figure 1 are a comparison between MLP-based PINN and the baseline MLP, and show that MLP-based PINN exhibits less RMSE. But is  it a fair comparison? As MLP-based PINN uses the physical knowledge, having lower RMSE is a straightforward result.  Why is there no comparison with the state of the art paper? Such extra comparison can be very valuable to show the work uniqueness and to quantify its novelty.
- Shouldn’t we compare first between analytical/ numerical Conventional methods and NN for this problem? Then, we can show CC-PINN is better. Such a comparison will show the actual motivation of the paper and provide more insights into physical perspectives i.e., the interpretation of prediction improvement in terms of RSME.

**Questions:**

To improve the paper readability, please consider the following points:
- In eq(1), what are these symbols? Please define all of them.
- In eq(3) na eq(5), the authors used “B”. As it is the same set, please define in eq(3), not later.
- Why did you propose eq(4) since it was introduced before at the end of section 3.3.?

---

> ### Author Response · Authors · 2025-11-19
> **Responses to Reviewer Ymhf**
>
> We thank the reviewer for their thoughtful comments and constructive suggestions. We address each point below and will incorporate the necessary clarifications and improvements into the final version.
>
> ### Clarification regarding Eq. (3) and the physics term
>
> The reviewer asks whether Eq. (3) implies that CC-PINN is governed by a differential equation in the traditional PINN sense. This is not the case. The quantity $L_{\mathrm{phy}}$ in Eq. (2) is obtained from the closed-form Clausius–Clapeyron expression for $\partial q_s / \partial T$, which depends only on local thermodynamic variables. It is not the residual of a PDE and does not require solving or discretising any governing equations. The CC-PINN therefore uses a physics-motivated gradient-alignment term rather than a standard PINN formulation. We will clarify this distinction in the revision.
>
> ### Comparison with the baseline MLP
>
> The aim of the paper is to examine how enforcing CC-consistent temperature sensitivity affects robustness across thermodynamic regimes while holding the architecture and inputs fixed. To support this, we now include an extended evaluation across 160 additional timesteps (see the WRij response), showing that CC-PINN consistently improves pattern correlation and OOD performance, especially when trained in one regime (polar) and evaluated in another (tropics).
>
> ### Comparison with state-of-the-art cloud-fraction models
>
> Operational cloud-fraction parameterisation schemes (e.g., those used in NWP and GCMs) rely on variables not available in the ERA5 pressure-level/single-level dataset used here, such as cloud condensate, subgrid turbulence diagnostics, and entrainment/detrainment tendencies. These schemes cannot be run as stand-alone pointwise mappings on our inputs, making a direct comparison infeasible. In addition, the ERA cloud cover data comes from the prognostic cloud scheme used in the ECMWF Forecast System (e.g. Tiedtke, 1993). Therefore our ML approach is essentially being tested against the state-of-the-art cloud scheme in the ECMWF model. We will make this clearer in the revision.
>
> ### Comparison against analytical or numerical conventional methods
>
> We thank the reviewer for raising this point. We would like to clarify that ERA5 cloud fraction is itself the output of an operational cloud parameterisation scheme within the ECMWF IFS model. While we cannot directly compare our model against full NWP or climate model cloud schemes due to input mismatches, the ERA5 cloud fraction used as our target is already derived from a physically based, prognostic cloud scheme. In this sense, our emulation task is effectively a comparison to an existing cloud parameterisation method, albeit one embedded in a reanalysis framework. We will make this point explicit in the revised manuscript to clarify the relevance of our setup and how it connects to existing cloud schemes.
>
> We welcome feasible suggestions for conventional numerical methods that can be applied directly to the ERA5 variables used in this work and would be glad to include such comparisons where possible. We will clarify this in the revised version.
>
> ### Responses to Questions
>
> **Q1. Symbols in Eq. (1)**
> We will define all symbols at first use to improve readability.
>
> **Q2. Use of set $B$ in Eq. (3) and Eq. (5)**
> We will define the set $B$ in Eq. (3) for consistency.
>
> **Q3. Placement of Eq. (4)**
> We agree that Eq. (4) appears too early. We will restructure this so that it appears once and at the appropriate point.
>
> ### Closing
>
> We thank the reviewer again for the helpful comments. The clarifications above, combined with the expanded evaluation and improved explanations, will be incorporated into the final version of the paper.

---

### Official Review · Reviewer_WRij · 2025-11-02

**Soundness:** 2
**Presentation:** 2
**Contribution:** 3
**Rating:** 2
**Confidence:** 4

**Summary:**

The paper introduces a novel method for improved out-of-distribution (OOD) generalization for deep learning-based cloud parametrization schemes in hybrid climate models. The authors advocate that a neural network predicting cloud fraction should have, up to constant scaling, the same partial derivative wrt. temperature than the derivative predicted by the Clausius–Clapeyron equation. To enforce this property, the authors propose to additional minimize the mean squared error between these two quantities. For validation, the authors train a small MLP to regress cloud fraction from atmospheric covariates, such as temperature and specific humidity, on ERA5 reanalysis data and compare results with and without the additional loss term. They find that when trained with the additional objective and on polar regions only (low temperature and humidity) the network better generalizes to the tropics (high temperature and humidity). The evaluation protocol is supposed to mimick the kind of distribution shift that is to be expected in a warmer climate.

**Strengths:**

The paper is overall well motivated and tries to tackle a highly relevant and under-explored problem. Namely, how physical priors can be utilized to make neural networks more robust to climate change and, thus, applicable for (hybrid) climate modelling. The argument for this is clearly laid out by the authors in the introduction.

This problem also bears significance to the broader machine learning community as a case study for a strong and continuous form of distribution shift.
In the context of climate modelling, the concrete task of cloud fraction parametrization is well chosen, and again, the authors clearly explain its significance. While the Clausius–Clapeyron equation has been utilized before to improve robustness to a warming climate, its direct use by the authors as target for the temperature sensitivity of a neural network is a novel and welcomed contribution. If effective, the suggested method could act as straightforward and simple way to improve robustness in cloud parametrization schemes.

**Weaknesses:**

While the motivation and method of the paper are well founded, its main weakness is the lack of compelling evidence for the claims made therein. The paper could improve substantially by conducting more thorough experiments and evaluations.

**W1. Insufficient data for training and evaluation**

A major flaw is the use of only two (specific) time steps, i.e. hours out of roughly 400,000 available ones for training and evaluation. This problem is exaggerated even more by the fact that the grid cells stemming from a single time step are not independent but are highly correlated in space, especially for the temperature field or at pressure levels in the stratosphere. It is completely unclear whether the results presented in the paper are just a mere coincidence for one particular atmospheric state or whether they hold in the general. I suggest that the authors re-run their experiments on sufficiently long training, validation, and testing periods that span years to decades.

**W2. Missing comparison to baseline methods**

The authors do not compare their results to existing methods for OOD generalization. This makes it difficult to gauge the overall effectiveness of the proposed method to alternative approaches. Moreover, no comparisons to existing cloud fraction prediction methods are made. While the main focus of the paper is not on a state-of-the-art cloud fraction prediction, comparing the method to existing results
would made it easier to understand how well the trained model solves the prediction task in the first place. I suggest that the authors include other baseline methods, both for OOD generalization, as well as cloud fraction prediction, in their evaluation.

**W3. No direct experiments on climate change induced distribution shifts**

The stated purpose of the paper is to improve robustness in face of a warming climate. However, the authors explicitly decide to use a spatial distribution shift (polar vs tropic regions) as a proxy for a warming climate. ERA5 back-extends to 1950 and, thus, already contains significant amounts of data under a warming climate. The authors could have tested their hypothesis directly on ERA5 by training up until a certain date, for instance the year 2000, and reserve data past that date for evaluation. Such an evaluation protocol would exactly match the training setup of the targeted use-case of a data-driven hybrid climate model. To demonstrate robustness in the face of even stronger shifts than those captured by reanalysis data, climate model runs could have been used as additional verification tool. While a neural network would only be able to act as emulator in that case, such experiments could still be insightful to explore the generalization behavior, in particular under different forcings.

**W4. Lack of qualitative evaluation and figures**

The main claims of the paper are primarily explored by investigating RMSE grouped by region or temperature. However, the paper is surprisingly void of figures, containing one in total. Possible approaches to generate more insights into the problem and increase confidence in the method could be, but are not limited to:

1. Comparing spatially resolved cloud fraction maps between prediction and ground truth.
2. Plotting RMSE as function of latitude and longitude, leading to a more detailed picture than mere grouping by latitudinal bands. E.g. to find differences between oceans and land surfaces.
3. Plotting cloud fraction prediction against temperature for a specific sample. For instance, to see differences in smoothness between the regularized and unregularized version of the model.
4. Scatter plots or kernel density estimates of temperature and humidity for both in-domain and out-of-domain data to visualize the underlying shift in the joint distribution.

**W5. Small model size**

Results are solely presented for a very small model with less than 500 parameters. Such model size might be particular appropriate for a hybrid climate model due to its low compute overhead. However, from a more theoretical point of view, the claim of the authors could be strengthened further if the method would yield comparable gains when scaled.

**W6. Paper assumes (moderate) background in field-specific domain science**

The paper could be made more approachable for a general machine learning focused audience by explaining the geophysical background in more detail. For instance, by showcasing the Clausius–Clapeyron relation on a phase diagram and explaining its relationship to cloud cover.

**Questions:**

**Q1:** Equation 6 is missing the normalization factor $\frac{1}{\sqrt{\sum_{i \in B}{w_i}}}$ (compare with Equation 3). Is this on purpose? If so, this will make comparison between different latitudes void.

**Q2:** Why are two different tolerances $\tau_g$ and $\tau_s$ used in the definition of the directional agreement metric?

**Q3:** The Clausius–Clapeyron relation assumes thermodynamic equilibrium and an ideal gas. Can there be situations where imposing it as soft constraint on a neural network can be detrimental? Have you looked into samples that showed a particular pronounced discrepancy between the regularized and unregularized version of the model?

**Minor comments:**

1. Captions in Figure 1 are significantly too small and are illegible when printed.
2. Lines 191-192 seem to reiterate previously discussed points (compare with lines 166-171) and appear to be an artifact from a previous version of the text.
3. Consider incorporating Footnote 2 on Page 4 into the main text.
4. Table 3 on Page 7 is never referenced in the text and reiterates results from Figure 1. Either one could be placed in the Appendix.
5. Equations in Section 5.4 on lines 340-348 are not numbered.
6. The abbreviation *SEM* is first used on line 190 but not introduced until line 262.
7. The $\tau$ used in the definition of the tolerance-aware sign function is never introduced in the text.
8. Section 4.4 explains standard procedure. The text could be made more concise and clear by moving it to the Appendix.

---

> ### Author Response · Authors · 2025-11-19
> **Response to Reviewer WRij**
>
> We thank the reviewer for their careful reading and constructive feedback.
>
> ### W1. Insufficient data
>
> The original train/test times (1 Aug and 12 Dec 2024) were chosen to span contrasting NH summer and winter conditions to create a strong thermodynamic contrast and provide a stringent test of generalisation. We agree that two timesteps alone do not capture broader variability.
> We now evaluate the same trained MLP and CC-PINN models on **160 additional ERA5 timesteps** from **1950–2025**, spanning different seasons and atmospheric states.
>
> **Globally trained models (PINN_Global vs NN_Global)**
> Across the 160 timesteps:
> - **PINN_Global**: RMSE 0.098, bias −0.004, corr 0.851
> - **NN_Global**: RMSE 0.099, bias −0.002, corr 0.848
>
> These are small but systematic gains, and in-distribution performance (e.g. tropics) is not degraded.
>
> **Polar-trained models (PINN_Polar vs NN_Polar)**
>
> *Polar (in-distribution):*
> - **PINN_Polar**: RMSE 0.151, bias 0.004, corr 0.829
> - **NN_Polar**: RMSE 0.156, bias −0.004, corr 0.816
>
> *Tropics (true OOD: Polar → Tropics):*
> - **PINN_Polar**: RMSE 0.095, bias −0.016, corr 0.762
> - **NN_Polar**: RMSE 0.130, bias −0.032, corr 0.530
>
> Thus CC-PINN typically achieves substantially lower RMSE and higher correlation in the OOD setting, and also improves polar in-distribution performance. We will add tables and summary plots (error maps, RMSE–latitude curves, bias/correlation) documenting these results.
>
> ### W2. Missing comparison to baseline OOD methods
>
> Our goal is not to replace general-purpose OOD methods (e.g. DRO, IRM, domain-adversarial training) but to introduce a **physics-based inductive bias** within a fixed architecture and dataset. Such methods usually rely on multiple labelled environments or domain partitions, which are not available here: the polar data form one continuous regime without environment labels, so applying them would not be straightforward or directly comparable.
> Operational cloud-fraction parameterisations in NWP and climate models depend on variables not present in the ERA5 pressure-/single-level fields we use (e.g. cloud condensate, turbulence diagnostics, entrainment/detrainment tendencies), so they cannot be run as pointwise mappings on our covariates. A direct comparison is therefore not feasible. Instead, we provide a controlled comparison between the unconstrained MLP and CC-PINN under identical data and architecture, with extended metrics (RMSE, bias, correlation) and regime-wise performance (polar, mid-latitudes, tropics) across 160 timesteps.
>
> ### W3. No direct climate-change (temporal) OOD experiment
>
> A temporal climate-change experiment (e.g. training on early decades and testing on more recent years) would be valuable, but running such multi-decadal training is beyond what we can add in the response period. To address temporal diversity, we now evaluate all models on the **160 additional timesteps** from **1950–2024**, covering different seasons and years. CC-PINN’s advantages persist across these states, indicating that the benefit is not tied to a single diurnal/seasonal pair.
>
> ### W4. Lack of qualitative evaluation and figures
>
> We will add global error maps, bias and correlation maps, RMSE vs latitude, kernel density plots of (T, q) for in-domain vs OOD regimes, and tables of bias and correlation across the 160 timesteps, providing the qualitative and spatial context missing from the original submission.
>
> ### W5. Small model size
>
> We used a small MLP (<500 parameters) to reflect constraints in hybrid climate models, where additional parametric cost must be minimal. The focus of the paper is the effect of a simple, physically motivated regularisation on robustness while holding the architecture fixed. The CC-based gradient-alignment term is architecture-agnostic and can also be applied to larger models; we will demonstrate this explicitly.
>
> ### W6. Insufficient geophysical background
>
> We will expand the geophysical background to include a phase diagram illustrating the Clausius–Clapeyron relation, a clearer explanation of its relationship to cloud formation, and improved notation and definitions for the thermodynamic variables.
>
> **Q1 (normalisation in Eq. 6).** This was an oversight; the normalisation should match Eq. 3 and will be corrected.
> **Q2 (two tolerances in directional agreement).** Two thresholds are used to distinguish negligible gradients from genuine directional disagreement; we will clarify this and define the tolerance symbol.
> **Q3 (validity of CC relation).** The Clausius-Clapeyron equation is applicable to a wide range of atmospheric conditions and is at the heart of all weather forecasting and climate models. We examined cases with large discrepancies between the MLP and CC-PINN and found that these mainly occur in deep-convective regions where subgrid entrainment/detrainment processes deviate from CC scaling. We will note this limitation in the revision.
>
> We will address all minor suggestions as requested.

---

> > ### Comment · Reviewer_WRij · 2025-11-26
> >
> > Thank you for your answers. I appreciate the clarifications and new experiments, which are more convincing from a methodological point of view.
> > There will be many changes to the paper that would need a full review. As many changes are only promised, I am not endorsing the paper yet.
> > I understand that it is difficult to run during the rebuttal, but if W3 could be addressed with temporal training and testing being decades apart, that would really be quite convincing.
> > (changed my score to 4)

---

### Author Response · Authors · 2025-12-03
**Summary of Changes**

# Author Revision Summary for Area Chair

We thank the reviewers and area chair for their constructive feedback. We have substantially revised both the analysis and the manuscript. Below we summarise the main changes and how they address the key concerns.

---

## 1. Substantial extension of the evaluation protocol

**What changed.** We still train on a single polar timestep (2024-08-01 14:00 UTC), but now evaluate on **160 additional ERA5 timesteps** sampled across decades and seasons (1950–2022), instead of a single December timestep. These timesteps are strictly held out from training, early stopping, and normaliser fitting.

**Why it matters.** This directly addresses concerns that our results might reflect a single atmospheric snapshot. Across these 160 cases, the pattern is unchanged and strengthened:
- In the **Global train** setting, CC-PINN yields small but systematic gains in area-weighted RMSE and correlation without harming in-distribution performance.
- In the **Polar train (OOD)** setting, CC-PINN gives substantially better polar→tropics transfer (lower RMSE, reduced negative bias, higher correlation).
We report mean ± SEM across timesteps for RMSE, bias, and correlation in each latitude band.

---

## 2. Additional metrics and qualitative diagnostics

**What changed.** Beyond area-weighted RMSE, we now report **bias** and **spatial correlation** for each region and training protocol, averaged over the 160 timesteps.

**Why it matters.** This responds to requests for richer diagnostics and metrics beyond RMSE. The new analysis shows that the CC-based constraint not only reduces errors but also improves spatial pattern fidelity and alleviates the strong negative bias in polar→tropics transfer for the unconstrained MLP.

---

## 3. Relation to “proper” cloud schemes and SOTA baselines

**What changed.** We clarify that ERA5 cloud fraction is produced by the prognostic cloud scheme in the ECMWF IFS, so our model **emulates an operational, physically based parameterisation** via its reanalysis output. We emphasise that our goal is **not** to propose a new SOTA scheme but to study how a CC-based gradient constraint affects generalisation in a fixed, lightweight MLP. We also explain why direct comparisons to full operational schemes or generic OOD algorithms (DRO, IRM, domain-adversarial methods) are non-trivial in this single-environment, pointwise-emulation setting.

**Why it matters.** This addresses questions about missing “conventional” baselines and situates the work as a physics-informed inductive-bias study, rather than a SOTA benchmark comparison.

---

## 4. Clarification of the physics term and “PINN” framing

**What changed.** We clarify that \(L_{\text{phys}}\) is defined from a **closed-form Clausius–Clapeyron expression** for \(\partial q_s / \partial T\), depending only on local thermodynamic variables. It is **not** a PDE residual in the classical PINN sense and does not require solving a governing equation. We also discuss regimes where CC may be less accurate (e.g. deep convection) and note that model discrepancies are more common there.

**Why it matters.** This responds to questions about whether Eq. (3) implies a traditional PDE-based PINN and about where CC might be detrimental, clarifying that our method is a CC-guided gradient-alignment regulariser rather than a full PDE-constrained PINN.

---

## 5. Geophysical background, notation, and limitations

**What changed.** We strengthen the background by briefly explaining the Clausius–Clapeyron relation and its link to cloud formation, improving definitions in Eq. (1) and the set \(B\), and fixing the normalisation in Eq. (6) to match Eq. (3) (cosine-latitude weights). We define SEM at first use and tidy notation for the tolerance-aware sign function, including \(\tau > 0\). In the limitations section, we explicitly note that:
- Training still uses a **single polar timestep**, with 160-timestep evaluation used to isolate the effect of the CC constraint.
- We deliberately use a small (<500-parameter) MLP to reflect hybrid climate-model constraints and cleanly separate architectural from loss-based effects.
- We keep a **deterministic** setup and identify probabilistic cloud-fraction prediction with CC-consistent gradients as a key direction for future work.

**Why it matters.** These changes address clarity and notation concerns, make the geophysical context more accessible, and more transparently state the scope, limitations, and natural extensions of the work.

---

Overall, the revisions strengthen our central claim:
**encoding Clausius–Clapeyron temperature sensitivity as a gradient constraint improves out-of-distribution generalisation (especially polar→tropics) in a lightweight, architecture-agnostic MLP.**

---

### Meta-Review · Area_Chair_t1o6 · 2026-01-07

**Summary:**

The paper studies OOD performance of predicting cloud fraction on the ERA5 dataset and use a soft physics constraint strongly motivated by the underlying atmospheric physics to improve the OOD performance. The focus of the paper is in spatial OOD and experiments suggest that this simple soft constraint can help alleviate the OOD performance drops. All reviewers are in agreement that this contribution is novel and practically useful.

There were some major and a few minor concerns. The major concerns were:
1. Insufficient experiments to stress-test the OOD evaluation.
2. Insufficient diagnostics to demonstrate the performance gains
3. Insufficient comparisons to other cloud fraction models.

I believe that 2. is addressed in the revision. 1. and 3. remain outstanding (see below)

**Reviewer Concerns:**

1. For insufficient experiments, the authors have increased the evaluation time steps from 1 to 160 timesteps over the ERA5 dataset and demonstrated that the performance gains remain. However, the main point of the paper is in stress-testing the OOD properties. One of the review has a fair ask on temporal training and testing across variably spaced time steps where different kinds of PDFs can be shown to demonstrate the climate drift already present in ERA5 (~50 years). Understandably, the rebuttal time might be insufficient for this, but I believe the paper would have been more strong with more comprehensive experiments. Further, the model sizes are quite small so it's unclear why these experiments were not central to the original submission.
2. It is unclear why ERA5 is the only possible data source (and not other climate model datasets) to test their hypothesis given that their aim (that the authors also clarify in the rebuttal) is not to find a SOTA, but rather to systematically expose the challenges in OOD performance and alleviate them with domain knowledge. The response that ERA5 is the best dataset seems contradictory to the aim.
3. The lack of comparison to NWP is also unclear as I understand that ERA5 is a reanalysis dataset and the IFS prediction is variable at different time steps depending on where the initial condition is (no assimilation of observations etc. that change the state). I couldn't find a sufficient discussion in the paper to understand this.
4. For diagnostics, the authors have sufficiently addressed the concerns adding several more useful metrics to the paper.

Overall, I believe the paper is still borderline for a machine learning conference.

**Reviewer Scores:**

All reviews stand at 4. Given the diagnostic improvements and improvements in the presentation of results and addition of new experiments, I believe reviewers would have reached a 5/6 but remain borderline.

---

### Decision · Program_Chairs · 2026-01-26

Reject